# A Super Anticorrosive Ultrathin Film by Restarting the Native Passive Film on 316L Stainless Steel

**DOI:** 10.3390/nano13020367

**Published:** 2023-01-16

**Authors:** Ying Ren, Yuchen Li, Zhenwei Kang, Xiaoke Zhang, Shaojun Wu, Jun Shen, Genshu Zhou

**Affiliations:** Center for Advancing Materials Performance from the Nanoscale (CAMP-Nano), State Key Laboratory for Mechanical Behavior of Materials, Xi’an Jiaotong University, Xi’an 710049, China

**Keywords:** ultrathin metal oxide film, corrosion resistance, passive film, low-temperature oxidation, 316L stainless steel

## Abstract

The corrosion resistance of stainless steel is attributed to the extraordinary protectiveness of the ultrathin native passive film (~3 nanometers) on alloy surface. This protectiveness, independent of alloying, can possibly be further increased by modifying the native film to resist corrosion in harsh conditions. However, the modification based on the film itself is extremely difficult due to its rapid, self-limiting growth. Here we present a strategy by using low-temperature plasma processing so as to follow the growth kinetics of the native film. The native oxide film is restarted and can uniformly grow up to ~15 nanometers in a self-limiting manner. High-resolution TEM found that the film exhibited a well-defined, chemical-ordering layered structure. The following corrosion tests revealed that the anodic current density of the alloy decreased by two orders of magnitude in 0.6 M NaCl solution with a remarkable increase of pitting potential. This enhancement is also observed in Fe-Cr alloys with Cr contents above ~10.5 wt.%. The superior protectiveness of the alloy is thus attributed to the continuous and thickened high-quality ultrathin Cr_2_O_3_ layer in the restarted film.

## 1. Introduction

Metals (except for gold) are naturally prone to passivation. A spontaneously formed ultrathin film (less than 5 nm in thickness) isolates the metal substrate from aggressive environments and protects metal from degradation [1]. It has been successful in modifying the native passive film via alloying to design anticorrosion alloys. However, it may exert negative effects on the bulk properties of alloys, such as mechanical properties. For example, excessive Cr concentrations will cause brittleness and cracking of alloys. Therefore, optimizing the structure or composition of the film is a preferred strategy for increasing the film’s protectiveness. Compared to traditional microthick coatings, ultrathin native films (less than 3 nm) with sufficient amounts of anticorrosion elements are able to effectively protect the stainless steel for a long time, even under harsh conditions. The outstanding protectiveness of passive film seems to be more dependent on the amount of anticorrosive elements in the alloy than the film thickness. Furthermore, some native films exhibit crack-free and pinhole-free, which is crucial to corrosion resistance [2,3,4]. Thus, if these native passive films are further optimized, it might be possible to create superior ultrathin barriers on alloy surfaces to resist severe corrosive environments.

Once a metal is exposed to the atmospheric environment, an indigenous oxide film will instantaneously form on the metal surface [2], which makes it impossible to modify the film during its growth. Some methods—such as photon-assisted oxidation [5,6] and ozone treatment [7]—have been developed for post-processing the native passive films, however, they exhibit limited effects on the corrosion resistance. Low-energy oxygen implantation on the bare metal surface (cleaned by Ar^+^ etching) has recently been reported as a promising method of promoting the growth of metal oxide films [8,9]. Nevertheless, the properties of those films were poorly studied due to the technical complexity of this method. In fact, the chemical composition and structure of the passive film on the metal substrate are determined by both the metal substrate and ambient atmosphere. Therefore, passive films formed in most terrestrial environments are composed of metal oxides (M_x_O_y_) with hydroxides or oxyhydroxides in the outer layer [10]. Inspired by the formation mechanism of native passive films, we have attempted to alter the atmosphere around metals in order to improve the naturally formed passive films. In the present work, a dielectric barrier discharge (DBD) was introduced as the source to generate a low-temperature plasma [11]. We found that if surface damage from ion bombardment could be avoided, the growth of native passive film would be promoted under room temperature and ambient pressure. What is more, this film growth is self-limiting like the native film rather than like traditional metal oxidation processes with parabolic kinetic growth, such as thermal oxidation treatment with a resulting micrometer-thick protective oxide film [12]. Thus, the native oxide film seems to be restarted (RNO) in the presence of the low-temperature plasma. Although the native film grew by several nanometers, this thin film exhibited superior corrosion resistance and long-term stability. A common 316L stainless steel (SS) is used as the substrate, as it is a widely used engineering material due to its good corrosion resistance.

## 2. Materials and Methods

### 2.1. Materials

Both 316L stainless steel (69.3 wt.% Fe, 16.38 wt.% Cr, 10.7 wt.% Ni, 2 wt.% Mo) and 410 martensitic SS (12.5 wt.% Cr, 0.30 wt.% Ni, Bal. Fe) were purchased from China Baosteel Co., Ltd. (Shanghai, China). Fe-Cr alloys with varying Cr content were homemade using a powder metallurgical method. Other pure metals, including Al (99.9%), Ti (99.5%), Cr (99.9%), and Fe (99.5%), are also treated with the same processing. Prior to the plasma treatment, 10 mm × 6 mm × 3 mm bars of bulk metals or alloys were ground and polished with 0.5 µm diamond. The samples were then ultrasonicated in ethanol for 15 min and subsequently cleaned using pure alcohol and distilled water. After cleaning, the samples were dried under compressed air and then placed in a dielectric barrier discharge (DBD) system (See Appendix A). To prevent the metal surface from damage caused by ion bombardment, the discharge gap between the parallel barriers was limited to 8 mm. A sinusoidal voltage was applied between electrodes by a high voltage generator. The input voltage was 30 to 50 V, and the input current was 1.0 to 2.0 A. The ambient atmosphere, with humidity ranging from 20% to 45%, was introduced into the system at room temperature. The temperatures of metal surfaces during processing were 80–110 °C. Treatment time, ranging from 10 min to 10 h, was varied to investigate the effects of corrosion behavior from the thin film.

### 2.2. Characterization

The electrochemical experiments were carried out using a potentiostat (VersaSTAT3F, Princeton, Oak Ridge, TN, USA) with a voltage sweep rate of 1 mV per second without deaerating. EIS measurements were performed using a sinusoidal wave input potential with an amplitude of ±10 mV ranging from 10 kHz to 10 mHz at an ambient temperature of 25 °C. A salt spray test with 0.85 M sodium chloride solution at 45 °C was employed to accelerate the corrosion rate of alloys according to ASTM. The experiment for each sample condition was repeated three times. A microscratch tester with a load of 10 N was used to expose an area covered by the native film on the treated alloy surface. The corrosion morphologies of 316L samples were examined using a scanning electron microscope (SU6600, Hitachi, Tokyo, Japan). The high-resolution transmission electron microscopy (HR-TEM) and high-angle annular dark field and scanning transmission electron microscopy (HAADF-STEM) images were collected using a TEM (JEM-F200, JEOL, Peabody, MA, USA) operating at 200 kV. The cross-sectional samples for ultrathin RNO films were prepared via the focused ion beam (FIB) lift-out technique using a Helios 600 FIB-SEM setup (FEI, Hillsboro, OR, USA).

X-ray photoelectron spectroscopy (XPS) using an energy analyzer (ESCALAB Xi+, Thermo Fisher, Waltham, MA, USA) with a focused monochromatic Al Kα radiation was applied to analyze the chemical composition of the metal surface. The basic pressure inside the analysis chamber was about 3.5 × 10^−9^ Torr. All samples prepared were loaded into the spectrometer immediately to prevent charge transfer loss. The experimental O 1s curves were deconvoluted using a mixed Gaussian and Lorentzian function (GL (80)). A smart-type background subtraction was performed before curve fitting [13]. The binding energies of elements were calibrated with respect to a neutral adventitious C 1s binding energy at 284.8 eV.

## 3. Results

### 3.1. Structure and Chemical Composition of Films

Since no variation in surface morphology can be observed via scanning electron microscopy or atomic force microscopy, it was inferred that the plasma exerted quite a mild influence on the steel surface. To further analyze the microstructure of the metal surface, TEM and HAADF-STEM coupled with energy-dispersive X-ray spectroscopy (EDS) were used to analyze the FIB lift-outs. Figure 1a shows a STEM-HADDF image of the passive film formed on RNO-1h sample (after 1 h of plasma treatment), revealing that about an ~8 nm thick film formed uniformly on the surface of 316L SS. It is worth noting that each metal oxide is continuous and straight, and the distribution of these oxides have typical layered structure, following the same sequence (Ni, Cr and Fe, from the inner to the outer) as that of the native passive film on 316L SS. [14] This layered structure has not been detected by aberration-corrected EDS mapping from the native film on the stainless steel, [14] mainly because the total thickness of the film was less than 3 nm. Meanwhile, the continuity obtained herein also suggested that each metal cation was enriched in one position of RNO film rather than dispersed like in the native film (the possible reason for this difference will be explained hereafter). More importantly, the thickness of Cr_2_O_3_ layer (2 to 3 nm) obviously increased after the processing compared to the layer in the native film. Additionally, the high-resolution TEM imaging revealed the amorphous structure of the RNO film (the same as the native film [14]). The interface between the substrate and the film was distinct and straight, and the crystal lattice on the substrate was clear (Figure 1c), further indicating that the treatment was mild and did not cause damage to the metal substrate. According to the low-magnification image (Figure 1b), the film is uniform and continuous over a range of approximately 700 nm, exhibiting certain roughness adaptability characteristics.

The chemical composition of the RNO film was further analyzed using X-ray photoelectron spectroscopy (XPS). The survey spectrum revealed that there were no new added elements in RNO-10h film (Appendix A). However, the whole spectrum shifted toward the lower binding energies, indicating that the film was negatively charged. For the 10 min sample, the Fe 2p core level spectrum at 706.5 eV representing the metallic state disappeared, meaning that the film had already thickened (Figure 2a). Meanwhile, Fe cations in the film seemed to be totally oxidized into Fe^3+^ ions, however the fitting results indicated that there was a small amount of Fe^2+^. (Appendix A) Although the chemical states did not vary with treatment time, its total content slightly increased. For Cr 2p spectra, the Cr^3+^ ions in the film were partly oxidized into Cr^6+^ (Figure 2b). Their relative content first decreased, possibly due to the increase of Fe^3+^, and then slightly increased with treatment time. The Cr 2p3/2 spectrum was further reconstructed using a series of five peaks for Cr^3+^ owing to multiplet splitting (Appendix A). [13,15] Both Ni elements in the inner layer and a small amount of Mo elements were also oxidized into high-valence cations (Figure 2c,d). However, XPS also revealed that some metastable high-valence cations could not be stabilized after a long time in the corrosive solution. For example, Cr^6+^ ions were reduced to Cr^3+^ ions after 3 d of immersion in 0.6 M NaCl solution at room temperature (Appendix A). It is noteworthy that the disappearance of metallic state and the further oxidation of metal cations on the metal surface after processing was a ubiquitous phenomenon observed not only on 316L SS but also on Al, Fe, Cr, and Ti surfaces (Appendix A).

### 3.2. Electrochemical Behavior and Long-Term Protectiveness

Figure 3a displays the dynamic polarization curves in 0.6 M NaCl solution at room temperature, revealing a notable reduction of anodic current density after treatment. The anodic current density decreased with increasing treatment time, and the pitting potential increased accordingly. For the alloy after 10 h of processing, the decrease in current density was two orders of magnitude relative to the bare 316L SS, and the pitting potential (~500 mV) increased by 400 mV, indicating that the susceptibility to pitting decreased to a large extent. Meanwhile, the small current spikes indicating the presence of metastable pitting near the pitting potential in the case of the bare 316L SS even disappeared after 1 h treatment or longer. The corrosion resistance was further verified by the potentiostatic polarization experiments at a potential below the pitting potential for ~100 mV (Figure 3b). It was found that after 1 h of processing, the alloy could maintain a low anodic current density of near 10 nA·cm^−2^ for 800 s, yet the bare alloy was resistant to corrosion for only 400 s due to the high and unstable anodic current density. EIS was also employed to evaluate the corrosion resistance of these thin films. In particular, the Nyquist plots in Figure 3c exhibit the same trend as the polarization curves, indicating that the resistance increased with increasing treatment time. The equivalent circuit (EEC) was used for fitting the above EIS data. Since two time constants were observed on bode plots of RNO-1h and RNO-10h, R_Ω_-(R_ct_//CPE_dl_) was applied for bare 316L SS and RNO-10min, while R_Ω_ (C_f_//R_f_-(R_ct_//CPE_dl_)) was chosen for RNO-1h and RNO-10h, respectively. This difference could be related to the film thickness, where the interfacial anodic reaction (R_ct_) due to electron tunneling was the rate-determining step for the former two, whereas the ionic diffusion in the film (R_f_) dominated for the latter two. According to the fitting results (Appendix A), the polarization resistance of 316L SS increased from 2 × 10^5^·Ω·cm^2^ to 4.8 × 10^7^·Ω·cm^2^ when the processing time was increased to 10 h, which was consistent with the above dynamic polarization results. The inset image in Figure 3b shows the optical micrograph of the sample after 1 h of processing. It is difficult to discern the surface changes with the naked eye. The long-term protectiveness of the RNO film was evaluated via a salt spray test (0.85 M sodium chloride, 40 °C). According to the SEM images in Figure 3e, after six months of exposure, the bare 316L SS was covered with some corrosion products on its surface. However, the surface of RNO 316L SS remained clean and flat, and scratches after polishing were still clearly visible, demonstrating the long-term stability of the RNO film. The notable differences in surface morphologies were also observed on the scratched RNO sample (1 M NaCl, 80 °C). Many areas on the scratch became dark after immersion for three weeks, indicating accelerated corrosion, while other areas covered with RNO film were still intact (see Figure 3f).

### 3.3. Effect of Cr Content on the Enhancement of Corrosion Resistance

According to the above experimental results, the highly enhanced corrosion performance was achieved in the case of the RNO film on 316L SS. Although metal cations were further oxidized after the processing, the long-term corrosion resistance could not be attributed to Cr^6+^ because of its reduction to the equilibrium state during the subsequent immersion in 0.6 M NaCl solution (Appendix A). Furthermore, the enhancement in corrosion resistance was also observed in the case of low Cr martensitic SS such as 410 SS. As shown in Figure 4a, the RNO alloy exhibited a similar polarization behavior to the RNO 316L SS in 0.6 M NaCl solution (Figure 3a), and the reduction in the anodic current density after 1 h of processing could reach nearly three orders of magnitude with a significant increase in the pitting potential. However, the enhancement could not be observed in the case of Fe-Cr alloys with Cr contents below ~10.5 wt.%, such as Fe-9Cr. In addition, the enhancement was not observed in 6% FeCl_3_ solution (not shown here), which is too corrosive for the RNO film. Therefore, it was suggested that the stability of the film related to chemical composition should be the precondition under harsh environments. The chemical layered structure as revealed by EDS mapping (Figure 1a) and Cr 2p XPS spectra indicated that the growth process could not enrich Cr cations in the RNO film. In a word, the enhancement of corrosion resistance associated to the critical Cr content suggested that the thickened continuous Cr_2_O_3_ layer (2–3 nm) on 316L SS might have originated from the already existing Cr_2_O_3_ layer in the native film (the thickness should be less than 2 nm, otherwise the layer should be observed by HR-EDS mapping), although this ultrathin layer in native film has not been observed directly using HR-EDS mapping. When the Cr content in the alloy is below the critical value, the rapid increase of FeO_x_ complexes (Figure 2a) may disrupt the distribution of CrO_x_ film and break their continuity during the subsequent treatment. Therefore, the treated Fe-Cr alloys with the Cr content below ~10.5 wt.% could not survive in 0.6 M NaCl solution due to the discontinuity of the Cr_2_O_3_ layer.

### 3.4. Improvement of Film Quality by Plasma Processing

As shown in Figure 4b, the enhanced corrosion resistance after 1 h of processing could be also found in Fe-Cr alloys with varying Cr contents (0 wt.%, 9 wt.%, 12 wt.%, and 17 wt.%) in borate buffer solution (weak corrodibility). In this electrolyte, the anodic current density barely depended on the Cr content but was entirely related to the quality of the modified film (note that anodic current density of pure Fe was also decreased by the treatment). On one hand, the further oxidation could have greatly decreased the anodic reaction rate at the film–electrolyte interface owing to the reduction in the metallic atom content in the native film since there were no XPS signals from metals after the processing (Figure 2 and Appendix A). On the other hand, the thickening of the oxide film could have further slowed down the diffusion rate of ions in the film, but the hindering effect was obviously limited. Given a notable decrease in the anodic current density in 0.6 M NaCl solution (Figure 3a) with increasing treatment time and the well-defined chemically ordered structure of the RNO film, the enhanced film quality might be associated with the decrease of defect amount because of the unique kinetic growth of such film. In addition, there were no changes in the transpassive potential. It is indicated that the anodic behavior of the RNO sample is closer to a native film passivated metal rather than a coating-enhanced one; for example, the distribution of potential drop on an electrode.

### 3.5. Electrically-Driven Growth of the Film at Low Temperature

The structure and quality of the RNO film are related to the growth kinetics. As mentioned above, the negative shift in the binding energy for RNO samples varied with treatment time and storage time, and one of the largest displacements (1.9 eV) was observed in a 10 h sample (Figure 5a). The negatively charged sample indicated that an electric field of ~1.3 × 10^6^ V cm^−1^ formed within the RNO film (1.9 eV divided by 15 nm according to [16]; the true electric strength should be greater than it because of the charge loss during transferring), which was very close to the critical electric field for inducing a further growth of the oxide film at low temperatures. [17] The process was conducted at room temperature under ambient conditions, and the temperature of alloy surface during processing was kept at 90–110 °C [11]. Meanwhile, the growth of the RNO film was self-limiting, similarly to the native film, as shown in Figure 5c. Therefore, the growth of the RNO film might have conformed to the Mott model of metal oxidation at low temperatures, according to which the growth of the native film is kinetically driven by a self-established electric field (Mott potential), as illustrated in Figure 5d. [18,19,20,21] In addition, this dramatic shift disappeared after the sample was washed with ultrapure water (Figure 5b), indicating that some species with negative charges adsorbing on the metal surface might lead to the electric field during processing. The main products generated by DBD could be O_2_^−^·(H_2_O)_n_, CO_3_^−^·(H_2_O)_n_, O_3_^−^ ·(H_2_O)_n_, NO_3_^−^, etc. [22] It is speculated that these active negative ions adsorb on metal surface, forming an electric field within the oxide film due to electrostatic induction and leading to the further oxidation of the alloy as well as the film growth. Moreover, the metal oxide film with the thickness of several nanometers usually exhibits defect-free features under electromigrative annealing due to the Mott potential. [4] Here, compared with the rapid growth of the native film, cations in the RNO film had a sufficient time for electromigration according to their respective migration rate during the film growth. Thus, the number of defects within the RNO film should be much lower than in the native film owing to the constant electric field on the metal surface.

## 4. Conclusions

Inspired by the native passive films on the metal surface, we successfully restart the growth of the passive film on the 316L SS surface via mild low-temperature plasma processing. The main conclusions can be drawn as follows: (1) The anodic current density of the modified 316L SS decreased by two orders of magnitude and the pitting potential increased by 400 mV in 0.6 M NaCl solution (25 °C). (2) The films with thicknesses of 5 to 15 nm were amorphous and possessed a well-defined layered and chemically ordered structure. (3) The enhanced corrosion resistance in 0.6 M NaCl solution could be achieved in Fe-Cr alloys with Cr contents above ~10.5 wt.%, indicating that the continuous Cr_2_O_3_ layer in the film is a prerequisite for protection under corrosive conditions. (4) Compared to the rapid growth of the native film, electromigrative annealing resulting from a slow self-limiting growth enabled one to hinder the formation of defects in the film, which was also attributed to the enhanced protectiveness. (5) The strategy proposed in this study can be extended to other metals and alloys, providing new respects in the fabrication of super-anticorrosive and high-quality ultrathin metal oxide barriers.

## Figures and Tables

**Figure 1 nanomaterials-13-00367-f001:**
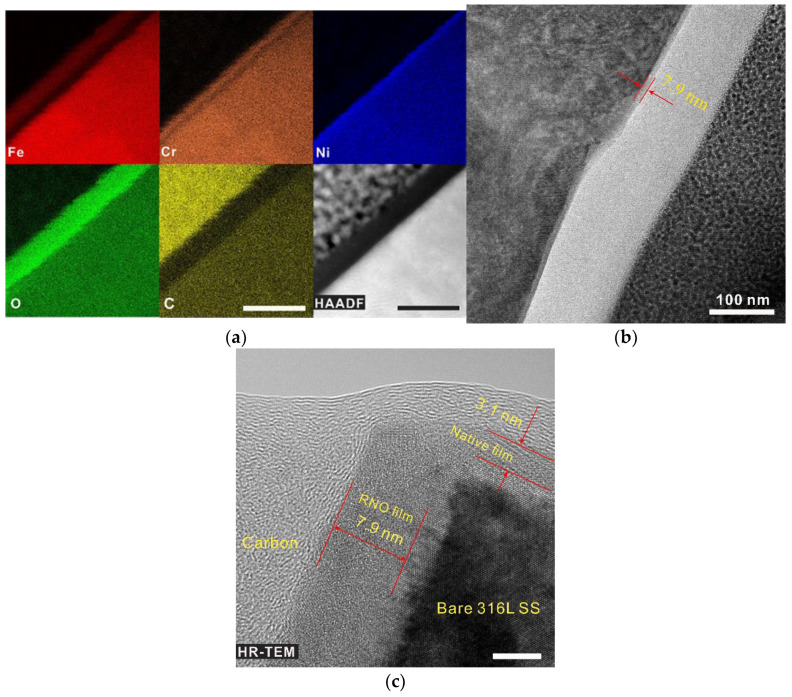
Structure of the RNO film. (**a**) High-angle annular dark-field (HAADF) scanning TEM image with EDS mapping of Fe, Cr, Ni, O, and C distributions within the cross-sectional RNO film on 316L SS after processing for 1 h; (**b**) TEM image of the alloy–film interface at low magnification. (**c**) High-resolution TEM image of the cross-sectional film. Scale bars (**a**), 20 nm, 5 nm.

**Figure 2 nanomaterials-13-00367-f002:**
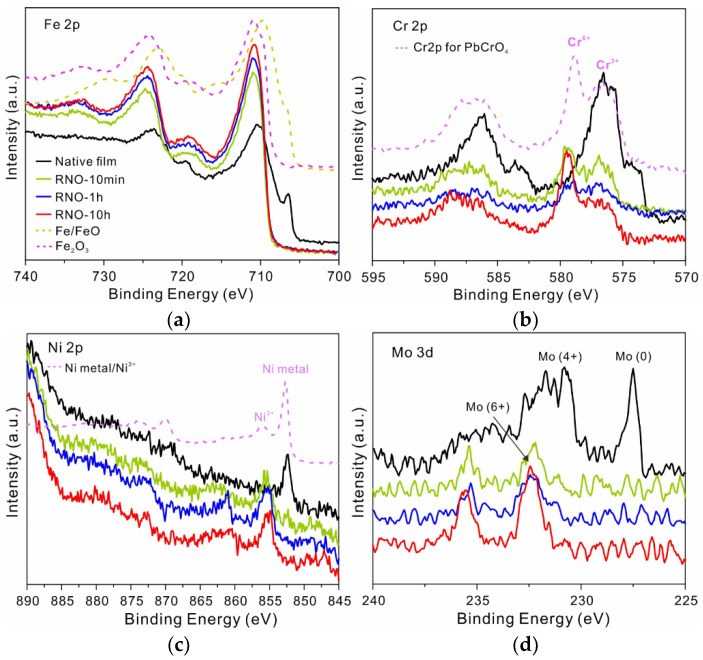
Chemical composition of the RNO film: (**a**) X-ray photoelectron spectra of the Fe 2p core levels of 316L SS films after DBD processing for various times and standard spectra of Fe^0^, Fe^2+^, and Fe^3+^ states for reference. (**b**) Cr 2p core levels of the films and standard spectra of Cr^3+^ and Cr^6+^ for reference. (**c**) Ni 2p core levels of the films and standard spectra of metal Ni and Ni^3+^ ions for reference. (**d**) Mo 3d core levels of the films.

**Figure 3 nanomaterials-13-00367-f003:**
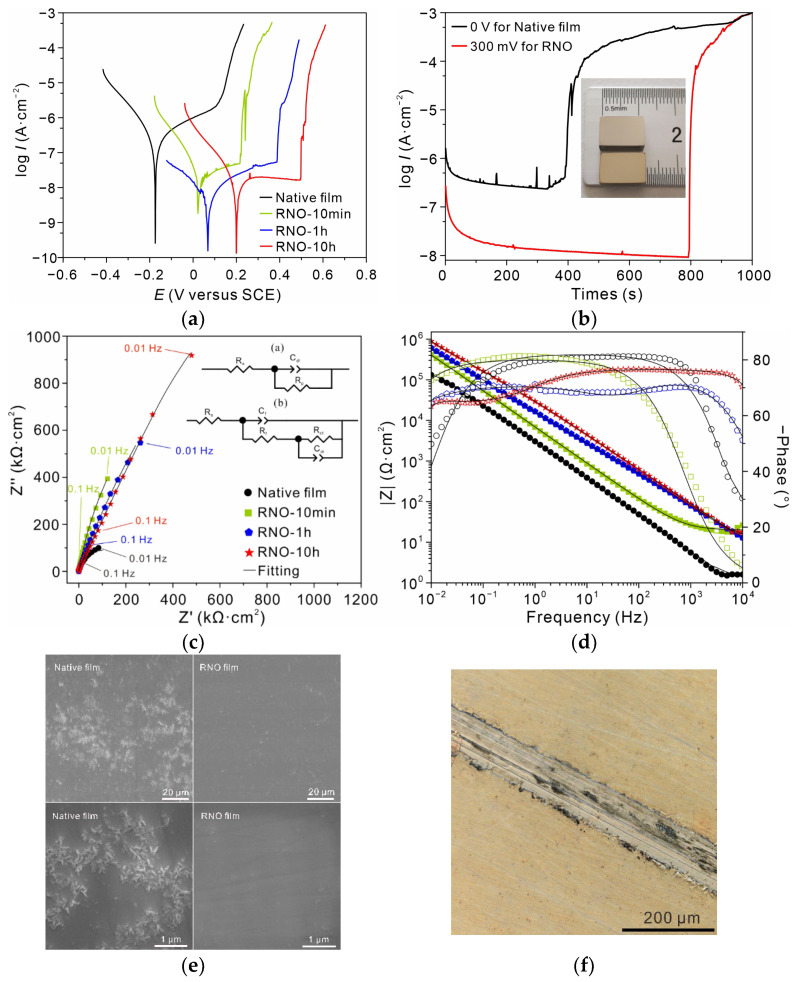
Corrosion electrochemical behavior of the RNO alloy. (**a**) Polarization curves of 316L SS after processing for 0 min, 10 min, 1 h, and 10 h in 0.6 M neutral NaCl solution at room temperature. (**b**) Potentiostatic polarization curves at 0 mV for bare 316L SS and at 300 mV for 1 h of processing, respectively. The inset is the optical photographs of 316L SS (top) and RNO-1h 316L SS (bottom). (**c**) Nyquist plots of the samples and the corresponding EECs used for fitting (solid line); (**a**) for bare 316L SS and RNO-10min; and (**b**) for RNO-1h and RNO-10h, respectively. (**d**) Bode plots and the corresponding fitting lines. (**e**) SEM surface morphology of bare (left) and RNO (right) alloys after six months of salt spray testing (40 °C). (**f**) Optical microscopy photograph of the scratched RNO alloy after 3 weeks of immersion in 1M neutral NaCl solution at 80 °C.

**Figure 4 nanomaterials-13-00367-f004:**
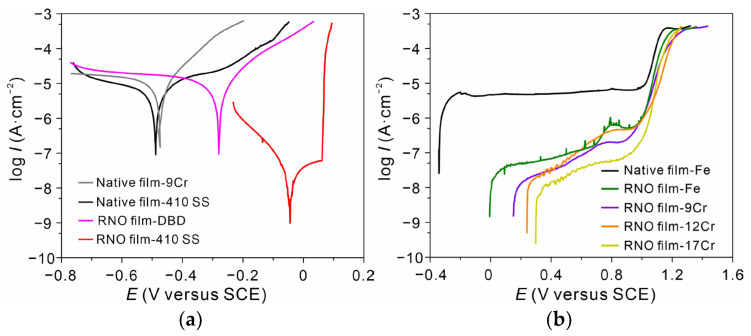
(**a**) Polarization curves of Fe-9Cr and 410 SS before and after 1 h of processing in 0.6 M NaCl solution (25 °C). (**b**) Anodic polarization curves of Fe-Cr alloys with varying Cr content in 0.2 M borate buffer solution (pH 7.8, 25 °C).

**Figure 5 nanomaterials-13-00367-f005:**
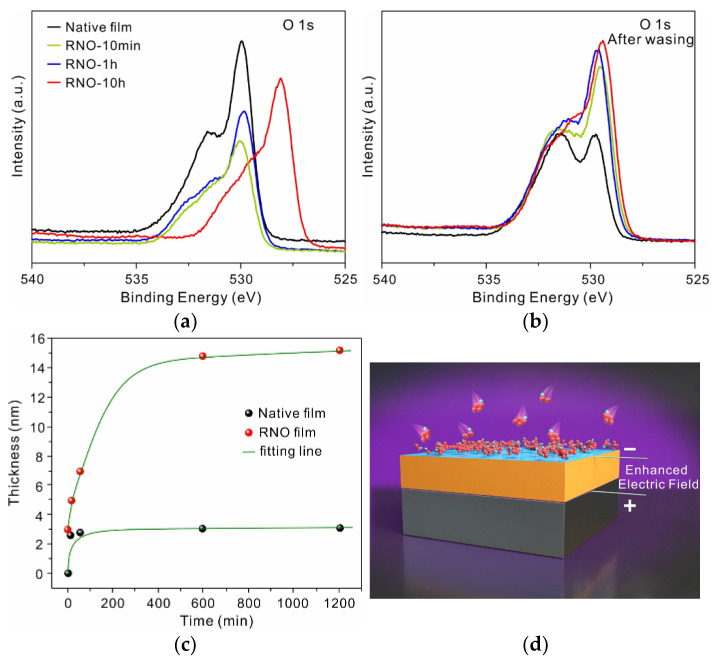
(**a**) XPS spectra of the O 1s core level in 316L SS films with varying treatment time. (**b**) O 1s spectra of the same samples after washing with ultrapure water. (**c**) Film thickness of 316L SS as a function of treatment time. (**d**) Schematic illustration of the growth mechanism of the RNO film.

## Data Availability

The datasets used and analyzed in the current study are available from the corresponding author on reasonable request.

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
