# Peer review of "A Super Anticorrosive Ultrathin Film by Restarting the Native Passive Film on 316L Stainless Steel"

_nanomaterials, 2023, doi:10.3390/nano13020367_

Round 1

Reviewer 1 Report

The authors claim to have created a super anticorrosive ultrathin film on stainless steel based on a well-defined layered and chemically ordered structure, but it is unclear to me how this layer is different from the passive layer that is formed in for example high temperature water. That the presence of a pre-oxidation passive layer improves the corrosion resistance is not new information and described in more detail elsewhere (e.g. Huang, X., Li, X., Zhan, Z. et al. Effect of Long-Term Pre-oxidation on the Corrosion Rate of 316L Stainless Steel in a High-Temperature Water Environment. J. of Materi Eng and Perform 31, 7935–7944 (2022). https://doi.org/10.1007/s11665-022-06878-2). This paper needs a good discussion on how this procedure relates to other corrosion methods (not only the native oxide layer) before it can be published.

Reviewer 2 Report

The article concerns the improvement of the protective properties of the native passive layer formed on steel as a result of low-temperature plasma treatment. The article is prepared well, but needs some corrections. In addition, in order to improve the quality of the article, the authors should be propose a mechanism for the growth of the layer thickness under the influence of plasma.

1. Can we talk about defects in the case of amorphous layers?

2. What does mean the claim "film exhibited a well-defined chemical-ordering layered structure"?

3. How should be understood the claim “that each metal cation could be much more concentrated than that in the native film”?

4. The fitting results in Table S1, in particular for Rct, are not logical. Therefore, statement 180-182"This difference could be related to the film thickness, where electron tunneling was the rate-determining step of the anodic reaction for the former two, whereas the ionic diffusion in the film dominated for the latter two." – may not be entirely true!

5. The authors wrote “The main products generated by DBD could be O2–, O4–, O3–, NO3–, etc. [20]". The authors of the work [20] did not specify this type of ions formed in the atmospheric plasma!

6. Can we say that the growth of the RNO film under the influence of atmospheric plasma can be consistent with ‘the Mott metal oxidation model at low temperatures, according to which the growth of the native film is kinetically driven by the intrinsic electric field (Mott potential)”. After all, the plasma process was for input voltage was 30 to 50 V, and the input current was 1.0 to 2.0 A.

Round 2

Reviewer 1 Report

The authors emphasized the difference with the high temperature water corrosion layer and the paper is fine for publication.

One small question: what is the origin of the carbon layer on top of the ultra thin film? Was it deposited during the FIB sample preparation or is it a result of the plasma treament?

Author Response

Carbon was deposited to offer protection for the film we fabricated during the FIB preparation.